# Role of sea ice, stratification, and near-inertial oscillations in

# 2 shaping the upper Siberian Arctic Ocean currents

3

1

- Igor V. Polyakov<sup>1</sup>, Andrey V. Pnyushkov<sup>2</sup>, Eddy C. Carmack<sup>3</sup>, Matthew Charette<sup>4</sup>,
- Kyoung-Ho Cho<sup>5</sup>, Steven Dykstra<sup>6</sup>, Jari Haapala<sup>7</sup>, Jinyoung Jung<sup>5</sup>, Lauren Kipp<sup>8</sup>, Eun
- Jin Yang<sup>5</sup>, Sergey Molodtsov<sup>2</sup>

- 1 International Arctic Research Center and College of Natural Science and Mathematics, University of
- Alaska Fairbanks, Fairbanks, 99775, USA
- 2 International Arctic Research Center, University of Alaska Fairbanks, Fairbanks, 99775, USA
- 3 Institute of Ocean Sciences, Fisheries and Oceans Canada, 9860 West Saanich Road, Sidney, BC, V8L 4B2, Canada
- 4 Woods Hole Oceanographic Institution, 266 Woods Hole Road, Woods Hole, Massachusetts, 02543, USA
- 5 Korea Polar Research Institute, Incheon, Korea
- 6 College of Fisheries and Ocean Sciences, University of Alaska Fairbanks, Fairbanks, USA
- 7 Finnish Meteorological Institute, Helsinki, Finland
- 8 Rowan University, 201 Mullica Hill Road, Glassboro, New Jersey, 08028, USA

Correspondence to: Igor V. Polyakov, 907-474-2686, ivpolyakov@alaska.edu

Abstract. The Siberian Arctic Ocean (SAO) is the largest integrator and redistributor of Siberian freshwater resources and acts to significantly influence the Arctic climate system. Moreover, the SAO is experiencing some of the most notable climate changes in the Arctic, and advection of anomalous Atlantic- (atlantification) and Pacific-origin (pacification) inflow waters and biota continue to play a major role in reshaping the SAO in recent decades. In this study, we use a large collection of mooring data to create a coherent picture of the spatiotemporal patterns and variability of currents and shear in the upper SAO during the past decade. Although there was no noticeable trend in the upper SAO's current speed and shear from 2013 to 2023, their seasonal cycle has significantly strengthened. The cycle reveals a strong relationship between upper ocean currents and their shear with sea ice conditions—particularly during transitional seasons—evidenced by a strong negative correlation (-0.94) between seasonal sea ice concentration and current shear. In the shallow (<20-30m) summer surface mixed layer, currents have increased because strong stratification prevents wind energy from propagating into the deeper layers. In this case, strong near-inertial currents account for more than half of the summertime current speed and shear. In the winter, a thicker surface layer is created by deep upper SAO ventilation due to atlantification, which distributes wind energy to far deeper (>100m) layers. These findings are critical to understanding the ramifications for mixing and halocline weakening, as well as the rate of atlantification in the region.

#### 1 Introduction

The Siberian Arctic Ocean (SAO), encompassing the Laptev and East Siberian seas and adjacent deep basins, plays a significant role in the Arctic climate system (**Fig. 1**). It is the Arctic Ocean's largest continental shelf system, and integrates Siberian freshwater inputs that account for over 10% of global river discharge. Moreover, the SAO is undergoing some of the most dramatic climatic changes in the Arctic (e.g., Polyakov et al., 2025a). Remote climate drivers, such as advection of anomalous Atlantic-(atlantification) and Pacific-origin (pacification) inflow waters and biota into the polar basins, have played a major role in reshaping the SAO. For instance, atlantification decreases stratification, enhances oceanic heat fluxes in the region, and has contributed to a decay of sea ice in the SAO over recent decades (Polyakov et al. 2023, 2025a). These physical changes have significant ecological consequences, including the arrival of new Pacific and Atlantic species in the East Siberian Sea (Ershova and Kosobokova, 2019).

**Figure 1**. Annual, summer (J–S), and winter (N–A) 2021–2023 mean current vectors and roses of instantaneous (hourly) currents for a depth range of 10–30 m at nine mooring sites in the Siberian Arctic. The mooring positions are represented by the beginning of the red vectors. Roses are placed to indicate exact directions.

Despite evidence of fundamental changes, such as regional reduction of sea ice, disappearance of permanent halocline, and altered freshwater transport across the SAO (e.g., Kipp et al. 2023, Polyakov et al. 2025a), quantification of many processes remains limited. At the same time, changes in upper ocean circulation due to climate change are not well documented—for example, to our knowledge, there have been no systematic current observations in the Makarov Basin. Moreover, the interaction between shortterm variability (from mesoscale to seasonal events) and long-term trends remains poorly understood. Particularly, the role of high-frequency processes acting at time scales of hours to days in rapidly changing SAO is not well understood whereas they are often the dominant components in time-series measurements of Arctic ice and ocean velocity variability (e.g., Plueddemann, 1992; Kwok et al., 2003; Rainville and Woodgate, 2009; Gimbert et al., 2012; Brenner et al. 2023a). These processes include semidiurnal tides and wind-forced near-inertial currents (NIC). Brenner et al. 2023a provide a good summary of near-inertial oscillations for further information. Details on the fundamentals of the mechanisms and drivers of the Arctic Ocean circulation can be found in (Timmermans and Marshall 2020). These knowledge gaps underscore the urgent need for in-depth analysis of the variability in the SAO currents at various time scales to better understand the dynamics and transformations of the SAO as it transitions to a new climatic state.

In this study, we build a cohesive picture of spatiotemporal patterns and variability of currents and shear in the upper SAO over the recent years using an extensive collection of mooring data. Shear plays a critical role in vertical mixing, stratification, and the distribution of heat and salt—processes that are especially important in the stratified and rapidly changing Arctic Ocean (e.g., Rainville and Woodgate 2009, Lenn et al. 2011, McPhee, 2008, 2013, Kwok and Morison 2017, Polyakov et al. 2017). Particularly, we map the time-dependent distribution of currents and their shear in the upper SAO; assess the role of sea ice, stratification, and NIC in setting the seasonal cycle of currents and shear in the region; and quantify the effect of winds on seasonal cycles of currents and shear in the surface mixed layer and halocline.

### 2 Data

Mooring observations: Our analysis utilizes the collection of instrumental observations of ocean currents from nine moorings distributed in the Siberian Arctic Ocean; see **Fig. 1** for locations. Moorings MB1 and MB3–MB9, maintained by the Nansen and Amundsen Basins Observational System (NABOS) program, were deployed from September 2021 through September 2023. Mooring KAMS1, maintained by the Korean Polar Research Institute (KOPRI), was deployed from August 2017 through September 2023. At

the MB1 mooring site observations began in August 2002, thus providing invaluable long-term measurements which were used to place shorter 2021–2023 records in a longer context. All moorings used in this analysis provided current observations from upward-looking 300-kHz Acoustic Doppler Current Profilers (ADCP) for the upper 40-50m. ADCP records were complemented in this analysis by profiles of currents from a McLane Moored Profiler (MMP) for the depths >50m at MB1 mooring. ADCPs provided current velocities, averaged over 2-m vertical cells, with at least 1-h time resolution. The manufacturer's estimates for 300-kHz ADCP accuracies are 0.5% of measured speed and 2° for current direction. The MMP sampled a vertical profile of water temperature, salinity, and current along the mooring line once per two days at a speed of  $\sim 25$  cm/s with a sampling period of 0.5 s; therefore, the data had a vertical spacing of ~12 cm. According to the manufacturer's manual, an instrumental error of the acoustic current meter (ACM) installed at MMPs is  $\pm 0.5$  cm s<sup>-1</sup>. The instrumental accuracy of the MMP magnetic compass is 2°. However, due to the weak horizontal component of the geomagnetic field in the Arctic Ocean, the individual compass error may exceed the instrumental accuracy, reaching 30° (Thurnherr et al., 2017); this issue similarly affects ADCP measurements. 2021–2023 CTD (Conductivity-Temperature-Depth) 48–1000m profiles from MMP and SBE-37 (34m) and RBR (21m) CTD time series from fixed depth observations at MB1 mooring were used in the study. The MMP temperature and conductivity calibrated measurement accuracies are  $\pm 0.002$  °C and ±0.002 mS/cm, respectively. SBE-37 and RBR provided 15-min interval records with measurement accuracies of  $\pm 0.002$  °C and  $\pm 0.003$  mS/cm for SBE-37 temperature and conductivity. All mooring data used in this study are available at https://arcticdata.io/catalog/#view/arctic-data. Winds: Monthly 10-m winds during 1979–2023 used in this study are from the European Centre for Medium-Range Weather Forecasts reanalysis ERA-5, downloaded from https://cds.climate.copernicus.eu/cdsapp#!/home (Herbach et al. 2023). The horizontal resolution of the data is 0.25°. Sea ice concentrations: The Advanced Very High-Resolution Radiometer (AVHRR) satellite archive used in this study includes global daily sea ice concentration from 1981-2023 with 0.25x0.25° spatial resolution (https://www.ncdc.noaa.gov/oisst). 3 Methods Defining buoyancy frequency (N): N is a measure of stability of a fluid. The mooring-based MMP measurements are used to quantify stratification using Brunt-Väisälä buoyancy frequency N,  $N^2 = -$ 

$(g/\rho_o)\partial\rho/\partial z$ , where  $\rho$  is the potential density of seawater,  $\rho_o$  is the reference density (1030 kg m<sup>-3</sup>), and g 116 is the acceleration due to gravity. Increased fluid stratification (stability) is indicated by increased N. 117 Mean current speeds: The mean current speed, |U|, for a specified time interval is obtained as the time 118 average of instantaneous speeds,  $|\mathbf{U}| = (u(t)^2 + v(t)^2)^{1/2}$ , where u and v are the east and north components of 119 either the total, measured current or the semidiurnal (near-inertial) band-passed current (see next 120 paragraph). 121 Vertical shear of horizontal currents: The vertical shear utilizing ADCP and MMP current observations 122 was calculated using the central finite difference method (when the shear at a vertical grid point is 123 proportional to the difference between the velocities of two nearby, shallower and deeper, grid points), 124 which produced a 4 m vertical scale for the shear estimates. 125 Estimating semidiurnal-band (near-inertial) and residual currents: Near-inertial currents were estimated 126 using hourly current zonal and meridional components that were band-pass filtered between 10- and 14-127 hour periods. The result of this procedure is referred to as NIC in the text. Residual currents are the result 128 of subtracting NIC from the original current records. 129 Selecting time intervals for comparative analysis of temporal changes: Our analysis builds on the 130 previous study by Polyakov et al. (2020a), which used the MB1 mooring records from 2004 to 2018. That 131 study identified a shift in the upper ocean current regime beginning with the 2013 record (noting a data 132 gap from 2010 to 2012) and provided comparative estimates of upper ocean currents for the periods 133 2004–2009 and 2013–2018. We extended the record by adding five more recent years, while maintaining 134 the same interval structure to ensure comparability of results. In the text, we included sensitivity tests by 135 reducing the latter period to 2018–2023 to match the number of observation years in the earlier interval, 136 which demonstrated the robustness of our estimates. We used slightly different peri 137 Simulating upper-ocean currents using a 1d model: The General Ocean Turbulence Model (GOTM; 138 Burchard et al., 1999; Umlauf and Burchard, 2005; Li et al., 2021) is a one-dimensional water-column 139 model that incorporates a library of turbulence closure schemes. It has been widely applied in open-ocean 140 studies (e.g., Burchard and Bolding, 2001; Le Clainche et al., 2010; Yu et al., 2022). In this study, we use 141 GOTM to simulate upper-ocean processes in the eastern Eurasian Basin of the Arctic Ocean at 78.5° N, 142 126° E. The model is configured with the Fairall et al. (1996) parameterization for air-sea heat fluxes, the

Winton et al. (2000) sea ice parameterization, and the  $k-\varepsilon$  turbulence closure scheme implemented with quasi-equilibrium second-moment closure.

GOTM is applied to resolve vertical velocity profiles and stratification dynamics of the upper Arctic Ocean, using both *in situ* profile observations and outputs from the E3SM-Arctic Earth System Model (Huo et al., 2025). Initial conditions are based on observed temperature and salinity profiles from January 2022 and September 2021, along with mean velocity profiles for January and June from the E3SM-Arctic "1950-control simulation" at the same location. Forcing parameters for winter/summer simulations include wind speeds of  $2.5 / 2.0 \text{ m s}^{-1}$ , surface air temperatures of -10 °C / -2 °C, ice thicknesses of 1 / 0 m, and shortwave radiation values of  $0 / 300 \text{ W m}^{-2}$ .

#### 4 Results

### 4.1 Documenting currents and shear in the upper Siberian Arctic Ocean in 2021–2023

The average currents for the upper 30m of the Siberian Arctic during a two-year period (September 2021 – September 2023) are shown in Fig. 1. Strong topographic steering causes currents in the outer western SAO (Eurasian Basin, MB1 and MB3 mooring records) to be along-slope even in the very top layer, supporting a previous finding by Pnyushkov et al. (2021). This SAO region has the strongest (up to 3 cm/s) mean currents, which do not vary greatly from season to season (Fig. 1). In the central SAO (moorings MB4–MB7), the currents become weaker (1-2 cm/s) and gentler slopes reduce topographic steering, and currents acquire a greater off-shore northward component, forming the origin of the Transpolar Drift (Kipp et al. 2023). The mean currents (vectors in Fig. 1) on the shelf (moorings MB8 and MB9) are weak because they lack a dominant direction, even if the instantaneous currents are typically greater, particularly in summer (as indicated by the red color of current roses). From 2021 to 2023, the East Siberian Sea shelf also shows a divergent pattern of seasonal currents, with winter currents predominantly flowing eastward (to the left according to the map orientation, MB8 and MB9 moorings) and summer currents flowing westward (MB9 mooring, with close to zero mean summertime flow at MB8 mooring location).

The hourly mooring records of total current speed  $|\mathbf{U}|$  and shear  $(\mathbf{U_z})$  in the upper 30m layer are shown in Fig. 2. The largest currents and shear correspond to the ice-free season in all mooring records, consistent with previous observations showing that compact sea ice cover dampens the upper ocean currents (Lenn et al., 2011; Lincoln et al., 2016; Rainville and Woodgate, 2009; Polyakov et al. 2020a). The spatiotemporal patterns of total and near-inertial currents and their shear during these two years are very similar, suggesting that the near-inertial currents play a key role in determining the dynamics of the

upper ocean in the SAO (**Fig. S1**). Annually and seasonally averaged total and near-inertial currents and their shear in the central SAO are shown in (**Fig. 3**). This figure provides further evidence that near-inertial currents contribute significantly to both total current speed and shear (on average 54% and 40%, respectively). This contribution is particularly significant in the summer, when, on average, the near-inertial component explained 64% (50%) of the total current speed (shear) in eight mooring records. This ratio was remarkably stable, geographically varying from only 69% (53%) for the annual records at the MB1 mooring site to 61% (56%) at MB9.

The annual and seasonal total and near-inertial currents were generally weaker in the central SAO than those in more eastern and western regions (**Fig. 3a-c**). For example, the total annual |U| minimum of ~ 4cm/s is found at the MB5 mooring location. Although the spatial pattern of shear is noisier, the MB5 mooring record still tends to show the weakest shear. At all SAO mooring locations, the shear in the upper 30m is significantly increased throughout the summer.

Figure 2: Biennial (September 2021 – September 2023) records, showing the magnitude of the total currents (left) and associated shear (right) at eight NABOS mooring locations (see Fig. 1) as a function of time and depth. White segments show missing data. Black-grey-white bar over each panel shows daily sea ice concentration between 0% (black indicating ice-free summer) to 100% (white indicating winter), with near grey scale for interseasonal transitions. Vertical dotted lines show transitions from year to year. Note that there are minor variations in the vertical data coverage of the mooring records.

Figure 3: Temporal changes in total (blue) and semidiumal-band (near-inertial, red) currents and their shear from September 2021 to September 2023 in the upper 30 m for eight locations in the Siberian Arctic. Estimates of (a, d) annual (January – December), (b, e) winter (November-July), and (c, f) summer (August-October) for mean current speed  $|\mathbf{U}|$  and vertical shear of horizontal currents  $|\mathbf{U}_z|$  respectively, at eight mooring locations. Statistical significance of means (error bars) is shown at the 95% confidence level.

M 88 M

\$ \$

# 4.2 Variability of currents and shear in the upper SAO

# 4.2.1 Seasonal variability in the upper SAO

The seasonal signal is the dominant component of Arctic atmospheric, sea ice, and oceanic variability. The winter winds, which are typically stronger than the summer winds (Fig. 4), should be the primary cause of this variability in the upper ocean currents. Indeed, surface-intensified upper ocean currents provide evidence that the atmospheric forcing dominates, particularly in summer (Fig. 2). At the same time, the ocean's response to this seasonal forcing is not straightforward due to a host of mechanisms and processes involved. For example, summer currents, not winter ones, are stronger in the upper 30m SAO due to (at least partially) the isolating effect of compact winter sea ice (Figs. 2,3 and 6).

**Figure 4**: (Top) 1958 –2023 time series of wind speed from monthly ERA-5 reanalysis for the M1 mooring location. (Bottom) Seasonal cycle of wind speed at the MB1 mooring location averaged over 1991–2006 (blue) and 2007–2022 (red). Shades represent 1 S.E. uncertainty. Note negligible trend and statistically insignificant temporal changes in seasonal cycles.

Figure 5: Time series of normalized (reduced to anomalies by subtracting means, Mn, and divided by standard deviations, SD) current speed  $|\mathbf{U}|$ , vertical shear of horizontal current  $\mathbf{U}_z$  (both from 10m depth level), and sea ice concentration (SIC, the latter time series are multiplied by minus one) at eight mooring locations. Blue lines show parameters for total minus near-inertial currents, red lines show parameters for near-inertial currents, and gray lines show SIC. Mn and SD are provided for  $|\mathbf{U}|$  in cm/s,  $\mathbf{U}_z$  in  $10^3$  s<sup>-1</sup> and SIC in %. Due to the large number of observations used in the analyses, the correlations R between  $|\mathbf{U}|$  and SIC (blue digits) and  $\mathbf{U}_z$  and SIC (red digits) are all statistically significant at the 0.05% level. This was determined using a standard Student's t-test, which assesses whether the observed correlation coefficients differ significantly from zero, accounting for sample size and assuming normally distributed

data. The effective number of degrees of freedom is computed following the Chelton's (1983) modified method which takes into account autocorrelations.

The connection between the sea ice state (expressed by its concentration) and the SAO's  $|\mathbf{U}|$  and  $\mathbf{U}_z$  is evident in Fig. 5, which shows an increase in currents and shear in the upper ocean (10 m depth) during summer associated with a decrease in sea ice concentration. In that, the start and end of summer intensification of |U| and, in particular, Uz at all SAO mooring sites closely follows the early summer sea ice decay and start of freezing in fall. This relationship is evident from the statistically significant correlations between the time series of sea ice concentration from one side and current speed and shear from the other side (Fig. 5). Statistical significance is estimated using a t-statistic, which follows a Student's t-distribution. The correlation analysis shows that sea ice concentration is generally more closely related to shear than to current speed (with the MB6 mooring as an exception), and that it is more strongly correlated with near-inertial currents than with residual currents (i.e., total minus near-inertial). Although the correlations between current speed and sea ice concentration are statistically significant owing to the long time series—they remain modest, reflecting the influence of other factors such as stratification and wind forcing on upper ocean dynamics. There is no obvious regional bias toward higher or lower correlation in the SAO. Seasonal cycles of sea ice concentration, |U|, and Uz averaged over longer (2013–2023) records from the MB1 mooring location further and more strongly corroborate this precise tuning of upper ocean dynamics and sea ice state (Fig. 6a,b).

# 4.2.2 Longer-term variability in the upper western SAO

Before discussing decadal-scale changes in stratification and currents in the SAO, we note that regional wind speeds have not increased, nor has the seasonal wind cycle undergone any statistically significant shift since the 1990s. Averaging over different time periods yielded practically the same results (**Fig. 4**). Therefore, other factors like sea ice and ocean stratification controlled the observed changes. In this analysis, decadal (2004–2023) records of total and near-inertial currents and shear from the MB1 mooring are used to validate results based on shorter 2021–2023 records and to provide further insight into interannual variability in the SAO.

The seasonal pattern of currents and shear in the upper 30m seen in MB1 2013–2023 mooring record (Fig. 7) is comparable to our results utilizing 2021–2023 mooring records from eight distributed mooring locations in the SAO (Figs. 2, 3). The strength of currents and shear in the eastern Eurasian Basin (MB1 mooring) during 2021–2023 were similar to those observed from 2013–2020, both annually and

seasonally, for total and near-inertial signals. For example, annual mean current speeds were 7.3 cm/s and 6.8 cm/s for 2013–2020 and 2021–2023, respectively; see **Fig. 8**. Throughout this decade, there was no apparent rising or decreasing trend in the current speed and shear of the upper ocean. Overall, the currents and shear were stronger than those seen from 2004 to 2009, notwithstanding some discernible interannual changes, especially during the summer (**Fig. 8**).

**Figure 6:** (a,b) Mean 2013–2023 seasonal cycle of current speed ( $|\mathbf{U}|$ , cm/s, blue), vertical shear of horizontal current ( $\mathbf{U}_z$ ,  $10^3$  s<sup>-1</sup>, red), and sea ice concentration (SIC, %, grey, multiplied by minus one) reduced to anomalies by subtracting means (Mn) and normalized by standard deviations (SD). (c-f) Mean 2004–2010 and 2013–2023 seasonal cycles of current speed ( $|\mathbf{U}|$ , cm/s, blue), vertical shear of horizontal

current ( $U_z$ ,  $10^3$  s<sup>-1</sup>, red), and SIC (%, grey). 10m depth at MB1 mooring location (eastern Eurasian Basin). Monthly running mean smoothing is applied to all time series.

**Figure 7**: September 2013 – September 2023 record of the total and semidiumal-band (near-inertial) current speed (cm/s) and associated shear (10<sup>3</sup> s<sup>-1</sup>) at MB1 mooring location (see **Fig. 1** for mooring position) as a function of time and depth. White segments show missing data.

**Figure 8**: Temporal changes in total (blue) and semidiurnal-band (near-inertial, red) currents and their shear from 2014 to 2023 in the upper 30 m. Estimates of (a, d) annual (January – December), (b, e) winter (November-July), and (c, f) summer (August-October) for mean current speed  $|\mathbf{U}|$  and vertical shear of horizontal currents  $|\mathbf{U}_z|$  respectively, at M1 mooring location. Dashed blue and red horizontal lines indicate 2004 –2009 means. Statistical significance of means (error bars) is shown at the 95% confidence level.

The late 2015–early 2018 period stands out due to its decreased currents and shear in every season. During this period, anomalously fresh water was advected from the Kara Sea through the Vilkitsky Strait (owing to amplified discharge from the Enisey River), causing a major freshening of the upper eastern Eurasian Basin (work in progress). Strong stratification in the area had a profound effect on upper ocean dynamics and restricted upward heat fluxes from the ocean interior (see Fig. 4 from Polyakov et al. 2020b). This may explain why throughout the summer months of these years, sea ice has been present in the area, but during the summer months of the previous and following years, there was none at all (Fig. S2).

A close examination of the eastern Eurasian Basin's decadal records of sea ice concentration, shear, and current speed reveals that the seasonal cycle has changed throughout time. In particular, summers associated with reduced sea ice concentration appear to have been longer in recent years across this decadal record, and the seasonal pattern of near-inertial and total currents has changed accordingly (**Figs.** 

7, S2). To place these changes in an even longer timeframe, we compare MB1 mooring records starting in 2004 to investigate the evolution of the seasonal cycle of sea ice concentration, |U| and  $U_z$  (Fig. 6c-f). The original current records from 2004 to 2010 used in our analysis are shown in Fig. 4 from Polyakov et al. 2020a.

When comparing seasonal cycles of sea ice concentration, |U| and  $U_z$  averaged over 2004–2010 and 2013–2023, a distinct and obvious tendency is the extension of the summer season for both sea ice concentration and currents. For example, the 80% sea ice concentration-defined winter-summer transition now occurs 20 days later in the fall and six days earlier in the spring (Fig. 6c-f). This is consistent with the findings by Stroeve and Notz (2018) which showed that in recent decades surface melt onset happens earlier and the freeze-up later. The currents and shear closely follow these seasonal changes in sea ice concentration. In the eastern Eurasian Basin, Polyakov et al. (2020a) found an increasing coupling between the vertical shear of oceanic currents, wind, and ice from 2004 to 2018, particularly during the summer. Extending this record by additional five years, we corroborate these earlier results. Specifically, we show that the seasonal cycle of  $|\mathbf{U}|$  and  $\mathbf{U}_z$  has strengthened, with larger differences between summer and winter—for example, the seasonal range of near-inertial currents increased from 4.8 cm/s (2004– 2010) to 8.0 cm/s (2013-2023). We also find a stronger negative correlation between seasonal cycles of sea ice concentration and current strength and shear in recent years. For example, the correlation between seasonal cycles of sea ice concentration and shear was R = -0.73 from 2004 to 2010, and a higher R = -0.94 from 2013 to 2023. This tightening of relationship indicates that currents and shear follow now closer the seasonal sea ice transitions than in the past. Reducing the latter period to 2018–2023 to match the number of observation years in the earlier interval yields a similarly high correlation of R = -0.89, indicating the robustness of our estimates.

#### 4.3 Role of stratification

The depth of the surface mixed layer in the Arctic varies seasonally, thickening in winter ( $\sim$ 25 to >50 m) and shallowing in summer ( $\sim$ 5–30 m), and regionally, with a deeper layer in the Eurasian Basin, ( $\sim$ 20 m in summer,  $\sim$ 70 to 100+ m in winter) than in Amerasian Basin ( $\sim$ 8 m in summer, 30 m in winter) (Peralta-Ferriz and Woodgate 2015). Since the SAO moorings lack temperature and salinity observations in the uppermost 20 m, we used mooring-based profiles of  $|\mathbf{U}|$  and  $\mathbf{U_z}$  to illustrate the seasonal evolution of upper ocean currents and their shear in relation to the stratification of the surface mixed layer (**Fig. 9**). This approach is supported by the relationship between upper ocean currents and stratification identified by Brenner et al. (2023b). During the winter months, residual currents and their shear show moderate

surface intensification, while the near-inertial component remains weak and nearly vertically homogeneous. In contrast, summertime currents are stronger—reaching 8–10 cm/s—dominated by the near-inertial component, and constrained by strong density gradients, with maximum shear located near the base of this stratified layer (e.g., Brenner et al. 2023b). This distinctive summer pattern of currents and shear can be understood as an indicator of stratification (see also **Fig. 7**).

**Figure 9**: Monthly mean 2004–2023 vertical profiles of (top) current speed and (bottom) vertical shear of horizontal currents. Total current speed and shear are shown in the left column, while semidiumal (near-inertial) current speed and shear are shown in the right. All data are from the MB1 eastern Eurasian Basin mooring.

The strong stratification-based bounding of upper ocean summer currents can be explained by the shallow water theory, which holds that the intensity of depth-averaged currents is proportional to wind stress and inversely proportional to layer thickness. Thus, thinner summertime surface layer has a higher concentration of wind energy and stronger currents. Stronger wintertime winds distribute energy across a thick, ventilated upper-ocean layer, leading to weaker currents (relative to summer currents) in the upper 30 meters—an effect influenced not only by sea ice shielding but also by changes in ocean stratification (e.g., Brenner et al. 2023a).

This interpretation is supported by observations in Fig. 10, which show total, near-inertial, and residual (total minus near-inertial) seasonal mean currents from 2018-2021 MB1 mooring record, averaged across 10-110m depth at 20m intervals. This limited period was selected due to the lack of high temporal resolution data from subsurface layers over a longer timeframe. Nevertheless, we argue that this two-year period is representative of longer-term conditions (e.g., Fig. 8) and sufficient to capture the major features discussed here. Expectedly, increased currents, dominated by the near-inertial component, are observed in the stratified upper 10-30 m layer in summer. At deeper depths (>30 m), a different pattern emerges: total and residual current speeds increase during winter—most probably driven by enhanced wintertime wind forcing—while no comparable intensification is observed in summer, and the near-inertial wintertime component remains weak. Thus, in summer, near-inertial currents drive a peak in average 10–110 m currents, whereas the wintertime peak is driven by stronger winds transferring energy through a deeper (>100 m) ventilated layer that extends into the halocline. Moreover, stratification is strongly connected to sea ice, and in the observed tight connection between the seasonal evolution of currents and sea ice concentration (discussed in section 4.2a) stratification is a physically significant intermediary link. Of course, this represents an average picture of the seasonal evolution of currents and shear in the upper SAO. In reality, each specific year and location introduces additional complexity, with numerous deviations in current behavior from the mean pattern (e.g., Fig. 5).

To verify that the effects observed in the observations are driven by seasonal variations in wind forcing, we conducted modeling experiments using the 1-D GOTM, forced with summer and winter wind conditions and realistic temperature and salinity profiles. The results clearly show the role of stratification in shaping upper-ocean currents, with maximum summer currents confined to the surface layer and deeper penetration of wind energy into the ocean interior in winter (Fig. S3). Additional experiments excluding sea ice confirm that stratification—rather than sea ice—plays the dominant role in the wintertime oceanic response to wind forcing.

Fig. 11 which shows the 2021–2023 mooring records from the upper 200m layer of the eastern Eurasian Basin, is used to further illustrate the points in the preceding paragraph. The profiler-based part of these observations (48–1000m) has a rather coarse (two-day) temporal resolution and cannot resolve near-inertial currents. However, they combine a relatively high vertical resolution of temperature, salinity, and currents. The temperature and salinity records clearly indicate deep (>150 m) seasonal ventilation, as shown by the cooling and freshening of the 150–200 m layer in late winter (February through May) in both 2022 and 2023 (Fig. 11a,b). The seasonal evolution of the upper ocean ventilation is reflected in deepening of strong shear at its base (upper 30 m) during the early winter period (Fig. 11e), followed by increasing buoyancy extending to ~80 m and deeper as winter progresses (Fig. 11f). By April–May, ventilation reaches well beyond 100 m, as indicated by the deepening of the buoyancy maximum to 120–140 m, which then persists throughout the summer months. This contrasts sharply with microstructure measurements from the ice-free Canada Basin during the 2012 'perfect storm,' which showed significantly enhanced turbulence in the upper 50 m of the water column. Despite this turbulence, Atlantic Water heat remained isolated beneath the strongly stratified halocline, across which thermohaline staircase structures persisted (Lincoln et al., 2016).

The connection between currents and stratification during the wintertime upper ocean ventilation is illustrated by their running correlation at 100 m depth, revealing a complex pattern of interactions of currents within the Arctic halocline (**Fig. 11c,d**). Notably, the correlation exhibits a clear and intriguing seasonal cycle: it is negative during summer and becomes strongly positive during the peak of the ventilation season and in the following months when stratification at this depth increases (**Fig. 11c**). The intensification of |**U**| and **U**<sub>z</sub> across strong density gradients at MB1 mooring site was documented by Polyakov et al. 2019; however, no physical explanation for the underlying drivers of this phenomenon was provided, and the mechanisms remain unclear. This may require advanced modeling efforts, potentially involving non-hydrostatic models.

Further evidence of the critical role ocean stratification plays in redistributing wind energy with depth can be found in **Fig. 6c,e**, which shows the presence of winter maximum of mean ocean current speed in the upper 30m layer due to stronger stratification and shallower mixed layer in 2004–2010, when ventilation in the upper Eurasian Basin was not that deep and stratification was stronger (Polyakov et al. 2017, 2020b).

**Figure 10**: (a,b) Mean 2018–2021 current speed averaged within 20m depth intervals (a) and over a 10 – 110m depth range (b). (c,d and e,f) Same as in (a,b) but for semidiurnal (c,d) and residual (total minus semidiurnal – e,f) currents. All data are from MB1 eastern Eurasian Basin mooring.

**Figure 11:** (a) Potential temperature (°C), (b) salinity, (c) running correlation (R) between current speed (|U|) and squared buoyancy frequency ( $N^2$ ) at 100 m depth using a 30-day running window (dashed lines: daily values; solid lines: smoothed using monthly window; statistically significant values at the 95% level are when |R| > 0.45, horizontal red segments show seasonal means), (d) wavelet transform of the R time series (95% significance and cones of influence indicated by grey lines), (e) vertical shear of horizontal currents (s<sup>-1</sup>), and (f) squared buoyancy frequency ( $10^{-5}$  s<sup>-2</sup>) in the eastern Eurasian Basin (MB1 mooring location) during 2021–2023. The black–gray–white bar above the top panel shows daily sea ice concentration, ranging from 0% (black) to 100% (white) on a linear color scale. White segments indicate missing data.

#### 5 Discussion

Mooring observations provided solid observational evidence that, predominantly wind driven, the upper ocean circulation in the SAO accumulates and converges flows of shelf freshwater from east and west in the East Siberian Sea, thereby acting as an integrator of river discharge (e.g., Jones 2001, Armitage et al., 2017). Cross-slope water and biogeochemical exchanges (e.g., Bauch et al. 2014), in contrast, are suppressed in the western SAO (Eurasian Basin) due to the constraining influence of strong topographic steering, such that flow, even in the very top layer, is aligned with the underlying topography regardless of wind direction (e.g., Pnyushkov et al. 2018). The mean currents in the western SAO region are the strongest, reaching up to 3 cm/s, and they vary little from season to season. Cross-shelf exchange is elevated east of the Lomonosov Ridge, where slope angles are relaxed. In this area, the currents become weaker (1-2 cm/s) and acquire a greater off-shore northward component, forming the origin of the Transpolar Drift. This freshwater pathway is confirmed by geochemical tracers such as radium-228, and high fractions of meteoric water (Kipp et al., 2023; Charette et al., 2020). Averaged over 2021–2023, the mean currents on the shelf are small because they lack a preferred direction, even if the instantaneous currents are typically greater than in the deeper ocean, particularly in summer.

From 2013 to 2023, there was no apparent trend in current speed or shear in the upper western SAO. However, both  $|\mathbf{U}|$  and  $\mathbf{U}_z$  were stronger during this recent decade compared to 2004–2009 (**Fig. 8**). This temporal pattern may be linked to alternating decadal-scale atmospheric variability (Polyakov et al., 2023). Over the past decade, the seasonal cycles of sea ice concentration, shear, and current speed in the upper SAO have also changed markedly. In particular, longer summers in recent years have led to earlier surface melt and later freeze-up. This has amplified the seasonal cycle of  $|\mathbf{U}|$  and  $\mathbf{U}_z$ —evidenced by greater summer—winter contrasts—and produced a close coupling between upper-ocean dynamics and sea ice conditions, as reflected by a strong negative correlation (R = -0.94) between the seasonal cycles of sea ice concentration and current shear from 2013 to 2023.

We confirm previous findings that currents in the halocline and surface layer are significantly impacted by the seasonal evolution of stratification. In the shallow (<20-30m) summer surface mixed layer, currents are increased because strong stratification constrains wind energy from propagating into the deeper layers. Strong near-inertial currents that are generated when sea ice is reduced account for more than half of the summertime current speed and shear. Published measurements of ocean turbulence indicate that high-frequency (like near-inertial and tidal) processes can dominate time-averaged mixing rates and diapycnal fluxes (e.g., Padman and Dillon, 1991; D'Asaro and Morison, 1992; Padman, 1995; Fer, 2014). In the winter, a thicker surface layer is created by deep upper SAO ventilation associated with

atlantification, which distributes wind energy to far deeper (>100m) layers. We provide evidence of winter amplification in the average currents across the 100m upper ocean layer in response to higher wintertime forcing. 1-D modeling experiments provide evidence that wind is the most likely candidate. Addressing this uncertainty may require more sophisticated experiments, potentially involving non-hydrostatic models. This winter amplification of currents and shear at depths is highly consistent with increasing seasonality and a general loss of double-diffusive staircases in the region over the 2004 to 2023 period (Lundberg and Polyakov 2025). Consequently, stratification represents a physically significant link in the strong connection between the seasonal evolution of currents and sea ice concentration. This is consistent with findings of Brenner et al. (2023a), who used a slab model results verified by mooring observations in the Beaufort Sea to show that both internal ice stresses and upper ocean stratification contribute to the intensity of inertial surface currents. Thus, while there has been no appreciable change in regional wind speed since the 1990s, sea ice and ocean stratification have been responsible for the observed changes in the upper SAO dynamics over the last few decades.

This analysis provides important insights on the consequences of ongoing atlantification of the SAO. For example, our finding of the dominant role of near-inertial currents in summer upper ocean dynamics over broad SAO area is critical for developing reliable mixing schemes for climate models. This would improve the model performance making the simulated future eastward progression of atlantification more certain. The identification of deep propagation of wind forcing in winter into the SAO interior is critical to understanding the ramifications for mixing and halocline weakening, as well as the rate of atlantification in the region. There are still numerous questions about the future of the atlantification and its impact on the short- and long-term physical, geochemical, and biological responses in the SAO and at pan-Arctic scales. As our mooring data demonstrate, extreme freshening of the upper ocean between 2015 and 2017 can in fact slow down the rate of atlantification. This is consistent with studies demonstrating that the recent freshening in the upper Canada Basin has enhanced the stratification regionally (e.g., Carmack et al. 2016, Proshutinsky et al. 2019, Timmermans et al. 2023, Haine et al. 2015) and may mitigate the consequences of atlantification. However, we do observe that atlantification is extending far beyond the Lomonosov Ridge, into the Makarov Basin of the Arctic Ocean with potentially profound implications for the regional ecological system (e.g., Polyakov et al. 2025a). Monitoring the fate of atlantification and understanding its role in climate change has great scientific and societal relevance, as this serves as a foundation for developing reliable forecasts and decision-making.

| 471 | Acknowledgements. IVP acknowledges funding from Office of Naval Research Grant N00014-21-1-                |
|-----|------------------------------------------------------------------------------------------------------------|
| 472 | 2577. IVP and AVP were supported by National Science Foundation (NSF) grant #1724523 and the U.S.          |
| 473 | Department of Energy grant 280253. LEK and MAC were supported by NSF grant numbers 2031853 and             |
| 474 | 2031854. KHC, JY, and EJY were supported by Korea Institute of Marine Science & Technology                 |
| 475 | Promotion (KIMST) grant funded by the Ministry of Oceans and Fisheries (KIMST RS-2021-KS211500,            |
| 476 | Korea-Arctic Ocean Warming and Response of Ecosystem, KOPRI).                                              |
| 477 | Competing interests: The authors declare that they have no conflict of interest.                           |
| 478 | Author Contributions. All authors participated in preliminary analysis; AP and IP carried out processing   |
| 479 | and analysis of mooring data, LK, MC, SD, JH and EC contributed to interpretation of hydrographic data and |
| 480 | formulating objectives of the study, KHC, JJ, and EJY provided processing and analysis of KOPRI data.      |
| 481 | All authors contributed to interpreting the data and writing the paper.                                    |
| 482 | Author Information. The authors declare no competing interests. Correspondence and requests should be      |
| 483 | addressed to IP (ivpolyakov@alaska.edu).                                                                   |
| 484 | Data Availability Statement. All mooring data used in this study are available at                          |
| 485 | https://arcticdata.io/catalog/#view/arctic-data. The ERA5 reanalysis data is available from                |
| 486 | https://cds.climate.copernicus.eu/cdsapp#!/home. Sea ice concentration is available from                   |
| 487 | https://www.ncdc.noaa.gov/oisst.                                                                           |
| 488 |                                                                                                            |

Armitage, T. W. K., Bacon, S., Ridout, A. L., Petty, A. A., Wolbach, S., and Tsamados, M.: Arctic Ocean 491 surface geostrophic circulation 2003–2014, The Cryosphere, 11, 1767–1780, 492 https://doi.org/10.5194/tc-11-1767-2017, 2017. 493 Bauch, D., Torres-Valdes, S., Polyakov, I., Novikhin, A., Dmitrenko, I., McKay, J., and Mix, A., 2014: 494 Halocline water modification and along slope advection at the Laptev Sea continental margin, 495 Ocean Sci., 10, 141-154, 2014 www.ocean-sci.net/10/141/2014/ doi:10.5194/os-10-141-2014. 496 Brenner, S., Thomson, J., Rainville, L., Crews, L. and Lee, C. M. Wind-driven motions of the ocean 497 mixed layer in the western Arctic. J. Phys. Oceanogr., 53, 1787-1804, doi: 10.1175/JPO-D-22-498 0112.1, 2023a. 499 Brenner, S., J. Thomson, L. Rainville, D. Torres, M. Doble, J. Wilkinson, and C. Lee, 2023: Acoustic 500 sensing of ocean mixed layer depth and temperature from uplooking ADCPs. J. Atmos. Oceanic 501 Technol., 40, 53-64, doi: 10.1175/JTECH-D-22-0055.1, 2023b, 502 Burchard, H., & Bolding, K. (2001). Comparative analysis of four second-moment turbulence closure 503 models for the oceanic mixed layer. Journal of Physical Oceanography, 31(8), 1943-1968. 504 Burchard, H., & Petersen, O. (1999). Models of turbulence in the marine environment—A comparative 505 study of two-equation turbulence models. Journal of Marine Systems, 21(1-4), 29-53. 506 Carmack, E. C., Yamamoto-Kawai, M., Haine, T. W. N., Bacon, S., Bluhm, B. A., Lique, C., Melling, H., 507 Polyakov, I. V., Straneo, F., Timmermans, M.-L.: Fresh water and its role in the Arctic marine 508 system: sources, delivery, disposition, storage, export, and physical and biogeochemical 509 consequences in the Arctic and global oceans. J. Geophys. Res. - Global Biogeochemical Cycles. 510 121, 675-717, doi:10.1002/2015JG003140/full, 2016. 511 Charette, M.A., Kipp, L.E., Jensen, L.T., Dabrowski, J.S., Whitmore, L.M., et al. The Transpolar Drift as 512 a Source of Riverine and Shelf-Derived Trace Elements to the Central Arctic Ocean. J. Geophys. 513 Res. Ocean. e2019JC015920. https://doi.org/10.1029/2019JC015920, 2020. 514 Chelton, D. B. (1983). Effects of sampling errors in statistical estimation." Journal of the Atmospheric and 515 Oceanic Technology, 2, 257–264. 516 Dasaro, E. A., and J. H. Morison, 1992: Internal waves and mixing in the Arctic-Ocean, Deep-Sea Research 517 A, 39(2A), S459-S484.

489

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

| 606 | Platov, E. Watanabe, T. Kikuchi, S. Nishino, M. Itoh, SH. Kang, KH. Cho, K. Tateyama, J.                     |
|-----|--------------------------------------------------------------------------------------------------------------|
| 607 | Zhao: Analysis of the Beaufort Gyre Freshwater Content in 2003-2018. J. Geophys. Res. 124, 9658-             |
| 608 | 9689, 2019.                                                                                                  |
| 609 | Rainville, L., and Woodgate, R. A.: Observations of internal wave generation in the seasonally ice-free      |
| 610 | Arctic. Geophysical Research Letters, 36, L23604. https://doi.org/10.1029/2009GL041291, 2009.                |
| 611 | Stroeve, J., Notz, D.: Changing state of Arctic sea ice across all seasons. Environ. Res. Lett., 13, 103001, |
| 612 | https://doi.org/10.1088/1748-9326/aade56, 2018.                                                              |
| 613 | Thurnherr, A.M., Goszczko, I., and Bahr, F.: Improving LADCP Velocity with External Heading, Pitch,          |
| 614 | and Roll, J. Atmos. Oceanic Technol., 34, 1713–1721, https://doi.org/10.1175/JTECH-D-16-0258.1,              |
| 615 | 2017.                                                                                                        |
| 616 | Timmermans, ML., J. M. Toole, The Arctic Ocean's Beaufort Gyre, Annual Rev. Mar. Sci., 15, 223-              |
| 617 | 248, https://doi.org/10.1146/annurev-marine-032122-012034, 2023.                                             |
| 618 | Timmermans, ML., and Marshall, J.: Understanding the Arctic Ocean circulation: A review of ocean             |
| 619 | dynamics in a changing climate. J. Geophys. Res.: Ocean, 125(4), 1-35                                        |
| 620 | https://doi.org/10.1029/2018JC014378, 2020.                                                                  |
| 621 | Umlauf, L., & Burchard, H. (2005). Second-order turbulence closure models for geophysical boundary           |
| 622 | layers. A review of recent work. Continental Shelf Research, 25(7-8), 795-827.                               |
| 623 | Winton, M. (2000). A reformulated three-layer sea ice model. Journal of atmospheric and oceanic              |
| 624 | technology, 17(4), 525-531.                                                                                  |
| 625 | Yu, C., Song, J., Li, S., & Li, S. (2022). On an improved second-moment closure model for Langmuir           |
| 626 | turbulence conditions and its application. Journal of Geophysical Research: Oceans, 127(5),                  |
| 627 | e2021JC018217.                                                                                               |
| 628 |                                                                                                              |