# Peer review of "Role of sea ice, stratification, and near-inertial oscillations in"

_EGUsphere, 2025_

## Author Comment (AC1)

**REVIEWERS' COMMENTS and ANSWERS**

**Reviewer #1**

**General comment**

My major concerns pertain to the clarity and novelty of the finding, rigour of the analysis and quality (and quantity) of illustrations.

As I understand it, the core message of the manuscript is that the seasonal cycle of speed (and shear) in the upper SAO is tightly linked to sea ice and has strengthened. The underlaying drivers are named (increased stratification in summer and increased ventilation during winter) but not shown in the manuscript (I understand that these are earlier findings by the same author(s)). Instead the focus appears to be on arguing that wind-driven near-inertial energy is distributed within the SML. This is in accordance with the over 50 year old concept of slab models, which the authors mention in passing but do not really incorporate in their analysis. Despite of the host of data presented, their argument remains largely qualitative and I find it difficult to decide how novel the insights actually are.

**A:** We sincerely thank the Reviewer for their careful reading of our manuscript and for the insightful comments provided. While we greatly appreciate the feedback, we respectfully disagree with what we perceive as a somewhat limited interpretation of our findings. As stated in the final paragraph of the Introduction, our objective was—and remains—to construct a cohesive picture of the spatiotemporal patterns and variability of currents and shear in the upper Siberian Arctic Ocean (SAO) over recent years, based on a comprehensive set of mooring data. We believe that the manuscript successfully meets this objective. To support this, we highlight below a list of key findings from our study, which we believe are both novel and quantitatively substantiated:

**a.** We document the spatial and temporal variability of currents and their vertical shear in the upper SAO, providing quantitative estimates such as means and standard deviations. These are novel results.

**b.** We assess the influence of sea ice on the seasonal cycle of currents and shear. While the broadening of the seasonal sea ice concentration (SIC) cycle is well established, our finding that upper-ocean currents and shear respond closely—and increasingly tightly over time—to changes in SIC is new and very important. This relationship is supported quantitatively by correlation analyses.

**c.** We quantified how winds influence the seasonal cycles of currents and shear in both the surface mixed layer (SML) and the halocline. Our results show that, in winter, weakened stratification due to Atlantification allows wind energy to penetrate much deeper than it could a decade ago—well beyond the SML. This leads to a wintertime peak in current speed and shear from deeper layers (quantified by providing estimates of means for current and shear). These are novel and significant results. The observed amplification of currents and shear below the SML during winter challenges the applicability of traditional slab models. This deep-reaching wind influence has significant observed and potential implications for upper ocean mixing and the regional marine ecosystem.

Thus, we believe our results are both novel and significant, and that they extend well beyond the more limited interpretation reflected in the Reviewer's comments. Nevertheless, we have taken all of the Reviewer's specific comments seriously and have addressed each one—either

by revising the text and figures accordingly or by providing a clear rationale for why certain suggested changes were not implemented.

**Specific comments:**

**SC1**: The only indication for a systematic change in seasonal cycles of currents, shear and sea ice is in the qualitative comparison of the 6-year average 2004-2010 with the 10-year average 2013-2023. In the time series presented, there is no obvious systematic change.

**A:** We respectfully disagree with this statement. In the manuscript, we provide quantitative statistical estimates of the seasonal evolution of sea ice concentration (SIC) over the past two decades (ll. 292-293 in the reviewed manuscript version). Fig. 7 clearly illustrates a strengthening of the seasonal cycle of upper ocean currents and shear, as evidenced by the greater range of seasonal fluctuations in panels 7c–f for the period 2013–2023 compared to 2004–2010. Additionally, the text (ll. 301-302) presents a quantitative measure—a statistically significant increase in the correlation between SIC and both currents and shear—demonstrating a tightening of this relationship over time.

**SC2**: The method introduced to separate wind-driven inertial currents from tides is not sufficiently explained, nor tested (but I also don't think it is necessary to attempt this separation in the first place).

**A:** We have removed this section from the manuscript, as it was not central to the objectives of our study.

**SC3**: There are far too many figures (over 100 panels in total) that are unfocused, hard to read and contain sometimes odd, unexplained choices. Most of them are hardly referred to and many do not clearly support the reasoning in the text.

**A:** We completely removed Figure 12 from the main text, moved Figure 11 to the Supplementary Materials, and entirely removed the former Figure S1.

**These are my short notes:**

**SN1**: Figures 1,2,3,4,6 all show seasonal cycle of upper ocean currents and shear in relation to sea ice. It should be possible to make this point with 2, max. 3 figures.

**A:** Fig. 1 does not show the seasonal evolution of currents in relation to sea ice; rather, it presents seasonal and annual mean current vectors and directional roses. Fig. 2 presents the original current data as a function of depth and time, while Fig. 3 displays their near-inertial component. These two figures form the basis for the subsequent analysis of currents in the upper SAO. Fig. 4 shows two-year averages of $|U|$ and $U_z$. Fig. 6, on the other hand, presents time series of $|U|$ and $U_z$ in relation to sea ice concentration. Therefore, we believe these figures are distinct and each serves a different purpose.

**SN2**: Figures 7-10 basically do the same for the 2014-2023 period.

**A:** We respectfully disagree. The focus of the first six figures is on spatial variability, whereas Figs. 7-10 emphasize temporal variations.

**SN3**: Figure 11 has no clear message; the increase of shear at the bottom of the SML is not really visible in most panels.

**A:** We removed this figure.

**SN4**: Figure 13 shows much of the same information as figure 12, but apparently with an inexplicable 2018-2021 time average.
**A:** We have removed panels a-d from this figure.

**SN5**: Figure 14: what is the physical motivation for the averaging times in c? d is the same as the top right panel of figure 2 and e is for some reason only given for a depth range from 60m, even though a and b go up to about 45m, which would be a very interesting depth for buoyancy frequency.
**A:** We revised this figure, eliminating completely panel c.
   Unfortunately, above 48m depth, we had two single-depth CTD sensors, whose data are sufficient for plotting temperature and salinity depth–time sections but are not suitable for estimating buoyancy frequency.

**SN6**: The list of references is not complete.
**A:** We revised the list of references.

**Suggestions for revision:**

**SR1**: Focus the manuscript on the core message of changing seasonal cycles from 2004 to 2023 (e.g. by showing full time series and deriving useful metrics).
**A:** We respectfully disagree with the reviewer's rather narrow interpretation of our objectives and main findings. As outlined in our responses above, their suggestion overlooks and unduly restricts the scope of our key results. We prefer to maintain the current focus and objectives of the study. Nevertheless, we have carefully addressed all of the reviewer's specific comments.

**SR2**: The point that there is little difference between all the MB moorings can be made much more succinctly.
**A:** The length of the manuscript is reasonable, and the result presented is both novel and significant.

**SR3**: Remove many of the figures (or at least put them into supplementary), revise and focus the remaining figures to directly and clearly support the argument.
**A:** We followed this advice and removed two figures (previously Figs. 10 and 11) from the main text. The current Figs. 10, 11, and 12 have been revised accordingly.
**SR4**: Perhaps add quantitative estimates using slab-models or at least incorporate the theory (and related findings) properly into the introduction.
**A:** We argued that slab models may be insufficient to simulate the complex processes observed in our mooring records. In particular, the wintertime amplification of currents and shear at the base of the ventilated layer—where strong vertical density gradients are present—may require sophisticated non-hydrostatic modeling experiments. However, we added some 1d modeling results emphasizing the role of wind and stratification in shaping upper ocean currents.

**SR5**: Decisions should be clearly motivated and transparent (why (and how) to use a 2-day window to separate tides from wind-driven currents, why the split at 2004-2010/2013-2023, why show 2018-2021 averages (fig. 13), why not show buoyancy in the 45-60m range (fig. 14)).
**A:** We removed the separation of tides and inertial currents from the text and added a section to the Methods to justify the time intervals we used. In the revised text, we also noted that the observations (formerly Fig. 14), obtained from single-depth instruments in the upper 50 m, are adequate for showing temperature and salinity, but are insufficient for estimating buoyancy.

**Line-by-line:**

**LBL1**: 23: Either expand on the "logistical challenges" or remove the sentence.
**A:** We removed this sentence. Thank you.

**LBL2**: 28: How can a cycle follow a coherence?
**A:** We edited this sentence.

**LBL3**: 29: Can the absence of ice "drive" something?
**A:** We edited this sentence.

**LBL4**: 54: Missing a "the".
**A:** We reviewed the sentence and found that it was missing a comma, which we added. However, we do not see where the article 'the' is missing in this sentence.

**LBL5**: 54-56: I think there should be references.
**A:** We added references, as requested.

**LBL6**: 56-58: Same here.
**A:** We revised this sentence to include a statement noting that, to our knowledge, there have been no observations of currents in the Makarov Basin of the Arctic Ocean. This type of statement is difficult to support with a citation.

**LBL7**: 60-61: No brackets necessary.
**A:** We believe that placing the references in brackets is appropriate in this context.

**LBL8**: 62-65: If this is to suggest a connection between inertial currents and Arctic Ocean circulation it should be explained here.
**A:** We believe that these two sentences are clear in meaning and do not require any changes.

**LNL9**: 69-71: References should be provided for this statement.
**A:** We added references as requested.

**LBL10**: 94: Is this no problem for the magnetic compasses in the ADCPs?
**A:** We revised this statement adding that ADCP measurements may be affected by the same issue.

**LBL11**: 95: Odd title for the section.
**A:** We modified this title.

**LBL12**: 123: I don't understand how this method could separates tides from wind-driven near-inertial oscillations. Where does the 2-day window come from? How is it implemented? What is the sensitivity? The explanation is lacking and the figure S1 does not provide useful information. In the context of this story, I think the authors could just proceed with NIC that contain wind driven and tidally driven currents (i.e. just do a 10-14h bandpass).
**A:** We removed this piece from the text.

**LBL13**: 153: Specify which moorings are considered to be on the shelf.
**A:** We added that MB8 and MB9 moorings are on shelf.

**LBL14**: 157: This is only (ever so slightly) visible for MB9.
**A:** We added clarification.

**LBL15**: 159-161: Sea-ice also provides a friction barrier for tidal currents.
**A:** We have revised the wording of this sentence—thank you.

**LBL16**: 192: The error bars appear to be exceedingly small; are they correct?
**A:** Yes, they are correct, given the number of observations used to compute each annual or seasonal mean.

**LBL17**: 203: Velocities in figure 2 mostly do not look surface intensified.
**A:** We added a clarification that the surface intensification is mostly in summer. Figure 12 offers a clearer visualization of surface-intensified currents. During the winter months, however, surface currents are only slightly stronger than those at deeper layers. Summer surface intensification is evident in Fig. 12. Thus, there is no contradiction in our statement.

**LBL18**: 221: Connection instead of connectivity?
**A:** We revised this sentence.

**LBL19**: 223: In the figure it is not possible to see if it is before, at the same time or after.
**A:** We edited the sentence.

**LBL20**: 225: How is the significance calculated?
**A:** We used a standard approach based on estimating t-statistics. We added a sentence in the text about that.

**LBL21**: 227: This does not appear to be the case for MB6.
**A:** We edited the text reflecting this fact. Thank you.

**LBL22**: 229: The differences often appear to be negligibly small.
**A:** The consistent nature of this relationship across most mooring records suggests that it is not a random occurrence.

**LBL23**: 234: Figure 7: Figure title talks of M1, should be MB1? Why split at 2010?
**A:** We renamed the mooring in Fig. 7. Figure 2 in Polyakov et al. (GRL, 2020) uses the same MB1 mooring records from 2004–2018 and clearly shows a shift in the upper ocean current regime beginning with the 2013 record (noting a data gap from 2010 to 2012). We extended the record with several recent years and found that the currents have remained relatively stable since 2013. This is the basis for our chosen separation. A description has been added to the Methods.

**LBL24**: 242: Wind speed is not the only meaningful variable; what about the wind direction or wind spectra?
**A:** Since most of our moorings are located in the deep, open ocean, the direction and shape of the wind speed spectra are less important in determining upper ocean currents.

**LBL25**: 247: In line 244 data from 2004-2023 is mentioned; why is only 2013-2023 considered here?
**A:** We partially addressed this in our response to LBL23. However, to compare the 2013–2023 data with earlier years, Fig. 9 includes the mean currents and shear from 2004–2009, shown as dashed lines.

**LBL26**: 254: Why is the data from 2004-2009 not shown?
**A:** They are shown as dashed lines in Fig. 9.

**LBL27**: 286: No obvious consistent change is visible in figure 10.
**A:** We observe that, in general, the summer minimum in sea ice concentration lasts longer in recent years, which broadens the summer peak of near-inertial (NI) currents. This is confirmed by the mean seasonal cycles shown in Figure 7.

**LBL28**: 314: This is hardly visible (if at all) in most panels of figure 11.
**A:** We removed this figure; instead a reference to Brenner et al. 2023b paper is included.

**LBL29**: 319: Should be Figure 13a-d.
**A:** Corrected, thank you.

**LBL30**: 322: Reference formatting.
**A:** We removed this reference from the text.

**LBL31**: 320-339: Isn't this the premise of all slab-models since about 1970?
**A:** That is correct. However, our findings extend well beyond the surface mixed layer, reaching depths greater than 100 m. Slab models—and ocean models in general—struggle to accurately capture the ocean's response to the complex interplay of wind forcing and stratification, particularly as it is influenced by Atlantification.

**LBL32**: 334-336: How are near-inertial wind signals "transformed" to non-inertial residual currents?
**A:** We revised the sentence to clarify that the near-inertial component refers to the currents, not the wind.

**LBL33**: 346: Why not show the full data set?
**A:** We have added the early years (see revised figure in the text), but there is little change in the monthly mean current profiles.

**LBL34**: 352: Where does the 2018-2021 average come from?
**A:** This analysis requires high temporal resolution current data. Such data are limited for the deeper layers (below the SML) by these years, therefore we used the available records to produce this figure. These years are representative of the last decade (e.g., Fig. 9). A detailed explanation is provided in the main text.

**LBL35**: 362-363: This is very confusing. Where am I supposed to see stratification decrease in winter? In figure 14c, buoyancy frequency tends to be highest in winter. Neither can I see greater currents or late winter shear in figure 14d.
**A:** We have revised this paragraph and updated Fig. 12 (previously Fig. 14). In particular, we now emphasize that temperature and salinity clearly indicate deep (>150 m) seasonal

ventilation, as shown by the cooling and freshening of the 150–200 m layer in late winter (February through May) in both 2022 and 2023. The seasonal evolution of the surface mixed layer (SML) can be traced by strong shear at its base (upper 30 m) during the early winter period (Fig. 12e), and elevated buoyancy extending to ~80 m later in the season (Fig. 12f). By April–May, the ventilation reaches significantly deeper, exceeding 100 m in depth.

The connection between currents and stratification is illustrated by their running correlation at 100 m depth, revealing a complex pattern of interactions within the Arctic halocline (Fig. 12c,d). Notably, the correlation exhibits a clear and intriguing seasonal cycle: it is negative during summer and becomes strongly positive during the peak of the ventilation season. We have added this discussion to the main text.

**LBL36**: 365: a and b are never discussed, it is unclear how the averaging periods in c are defined; why is buoyancy only shown from 60m, when T and S go up to 45m?
**A:** We edited this paragraph, incorporating references to Fig. 12a,b. Now, instead of averaging we used wavelet to show seasonality of interactions between stratification and currents.
In the text, we added a description noting that continuous, high vertical resolution MMP-based profiles of water temperature and salinity—suitable for calculating buoyancy frequency—were limited to depths below 48 m. Above this depth, we had two single-depth CTD sensors, whose data are sufficient for plotting temperature and salinity depth–time sections but are not suitable for estimating buoyancy frequency.

**LBL37**: 395: This is not visible in figures 9 and 10.
**A:** The mean 2004–2009 currents and shear are shown as dashed lines in Fig. 9, so this statement can be easily verified using these materials.

**LBL38**: 402: Where and how is this shown?
**A:** We added discussion to the text and updated Fig. 12 to include the correlation between current speed and stratification.

**LBL39**: 410-412: This is not shown.
**A:** We revised the text to reflect the uncertainty surrounding the nature of the winter forcing. Additionally, we noted that addressing this uncertainty may require sophisticated model experiments beyond slab-based approaches.

**LBL40**: 421-423: Is this new?
**A:** We revised the sentence to emphasize the new findings.

---

## Author Comment (AC2)

**REVIEWERS' COMMENTS and ANSWERS**

**Reviewer #2**

**Summary**
This manuscript investigates how seasonal shifts in sea-ice concentration and vertical density stratification regulate current speed and shear across the Laptev and East Siberian seas. The authors present an extensive mooring data set and deliver a largely qualitative interpretation of upper-ocean variability, highlighting the summertime prominence of near-inertial currents. Although the analysis is descriptive rather than quantitative, the topic is important and the observations are valuable for understanding ongoing Atlantification of the Siberian Arctic.

**Major Comments**

**Q1**: Several points require attention. Methodology needs greater transparency: lines 76–94 and 95–99 list instruments and programs, yet direct links or citations to the underlying mooring, sea-ice and wind datasets appear only later in the Data-Availability statement, which can be confusing. The ERA5 winds should be referenced in full (e.g. Hersbach et al. 2023, DOI 10.24381/cds.adbb2d47) rather than cited only by a portal link.
**A:** We added requested information to the text. Thank you.

**Q2**: The procedure described in lines 119–133 is understandable in outline, but additional detail is essential. There are no citations to the earlier works of Pnyushkov & Polyakov (2012) or Baumann et al. (2020, 2022); one must therefore assume previous approaches relied on Fourier transforms, whereas the present study may be using a wavelet-style filter to isolate the 12–14 h signal band. It is unclear what the declared "2-day window for NIC detection" actually means, given that NICs are already obtained through band-pass filtering. The 2012 paper referenced does not discuss spectral or frequency analysis, only time- and space-normalised diagnostics, so perhaps the authors intended the sliding-window harmonic analysis introduced in the Arctic Tidal Current Atlas by Till M. Baumann. The manuscript should explain why a 2-day window was selected, how window width influences the results, and how sensitive the analysis is to this choice.
**A:** Since both reviewers provided critical comments regarding this material and the topic is not central to our discussion, we have completely removed it from the manuscript and now focus exclusively on near-inertial currents.

**Q3**: A number of references cited in the main text do not appear in the

bibliography, for example More and Polyakov 2025, as well as the methodological papers mentioned above.  Completing the reference list will help readers trace the provenance of data and techniques.

**A:** We have revised the list of references – thank you for pointing that out!

**Q4**: Finally, the presentation could be streamlined.  The manuscript contains many multi-panel figures, some of which repeat similar seasonal or depth–time information.  Reducing the total number, improving resolution, and tightening captions would strengthen the flow and keep focus on the principal findings.

**A:** We have reduced the number of figures to 12 and revised the figure captions.

---

## Author Response (AR2)

**EDITOR AND REVIEWER'S COMMENTS and ANSWERS**

**Editor**

**Public justification (visible to the public if the article is accepted and published):**

The revised version of your manuscript has been substantially improved but there is still a number of issues, related to description of methods (calculation of errors) and visualization (too many and too complex panels) that need to be resolved. Therefore, the manuscript can be published subject to minor revisions as recommended by the referee.

Additional private note (visible to authors and reviewers only): Dear Co-authors,

The revised version of your manuscript has been substantially improved but one of the reviewers still have a number of remarks that must be taken into account. In particular, the way of calculating errors and the uncertainties in the correlations should be properly addressed. A few panels or figures that are not directly discussed in the manuscript could be moved to the supplement to enhance the clarity and readability of the paper.

These remarks are rather minor, and I hope that the next revision will be the final one.

Kind regards, Agnieszka

**A:** We have addressed all of the reviewer's comments. In particular, we expanded the description of our error calculation methods and reduced the number of figures in the main text.

**Reviewer #1**

**General comment**

I appreciate the authors' extensive response to my previous comments. I can see that in my attempt to distill the core message of the paper (and its vast amount of panels), I was too narrow. I appreciate the authors' explanation and their efforts to remedy some of the points I raised. At the same time, I still believe there is room for improvement in the manuscript and the illustrations.

Note: line numbers are based on the tracked changes version. (As a side note: Since the tracking of changes apparently did not work consistently, it is difficult to know how much the authors actually changed in the manuscript...)

**A:** We sincerely thank the Reviewer for their careful reading of our manuscript and for the insightful comments provided.

**Major Comments:**

**MC1:** The choices of calculating and presenting statistical confidence levels is unclear. Significance testing of the correlations is not explained/shown, apart from the mention that it involves a t-test. It seems odd that correlations as low as -0.19 are significant in this context (figure 6). Did the authors correct for autocorrelation in the time series (Chelton modified method)? How high does a correlation have to be to be above the confidence level in figure 6 (in figure 12, the level of 0.45 is given)? Revising the confidence intervals of the correlations may have ramifications for the conclusions drawn from figure 6.

**A:** We re-calculated the cross-correlations using the suggested approach, but the results remain unchanged: even the lowest correlation, R=-0.19, is statistically significant at the 0.05 level (p=0.002253). An explanation has been added to the text.

**MC2:** The "error bars" in figure 9 are all collapsed onto a point. Clearly this is not a useful way to display variability/uncertainty around the averages. Perhaps the standard deviation would be more useful than standard error in this case.

**A:** For the error bars, we used the same scale as defined by the vertical axes for the mean values. Due to the large number of observations used in calculating the means, the standard errors are very small. This is reflected in the narrow width of the whiskers in Figure 9. We are not aware of a clearer way to illustrate that the errors are minimal and the means are well defined.

**MC3:** In their effort to make a persuading argument, the authors at times use strong wording that does not appear to be supported by the evidence referred to. In lines 312-316, the authors claim that an increase in correlation from -0.73 to -0.94 (or -0.89, when using similar periods) shows a "tightening of relationship", with currents and shear following now "much" closer the seasonal cycle of sea ice. I am sceptical that this subtle change of correlation coefficients can be used to indicate a physical change of processes and it certainly does not support the strong wording used by the authors.

**A:** We have softened the tone of our presentation.

**MC4:** Along the same line: While running correlations in figure 12c seem indeed to be generally larger in late winter, there is no sign that they are (consistently and significantly) negative during summer (as claimed in line 385).

**A:** We added seasonal means, which clearly demonstrate that the sign of the correlations changes from one season to another.

**Minor comments:**

The authors still show figures with a great many sub-panels and details that are hardly (if at all) referred to or discussed. In particular, figures 2 and 3 are still only there to get a general impression of increased currents during the summer, that are (sometimes) gradually deepening. The difference between moorings is not really discussed in the text. So at least figure 3 could for example be moved to supplementary without loss to the story since quantitative differences between total and near-inertial currents are much easier seen in figure 4.

**A:** We moved Fig. 3 to Supplementary.

Figure 6 is very complicated, with lots of details and measures showing great variability between years and moorings. None of these measures or variability are discussed in the manuscript (there is one sentence in the discussion acknowledging that there is variability in figure 6, line 358). On the other hand, some of the same measures (correlations) done for

figure 7 are discussed in the text, but not shown in the figure.

**A:** Nearly the entire paragraph (lines 230–240) is dedicated to discussing this figure; therefore, we prefer to retain it in the main text.

**Line-by-line:**

129: Why is the wind time series in figure 5 split at 2006? That does not fit into the story at all.

**A:** We tested different time periods and obtained practically the same result; a comment has been added in the text.

235-246: It varies a lot in how far this simple relationship is visible at the different moorings and at different years. This is not mentioned at all. Also, correlations between -0.19 and -0.71 for total currents indicate that this relationship may be more subtle.

**A:** We added a sentence reflecting this fact. Thank you.

314: A modest increase in correlation does not indicate a "much" closer relationship. Consider adding the correlation values to figure 7 (as in figure 6).

**A:** We revised the wording of this sentence. Due to limited space in Fig. 7c, we decided it is sufficient to include these estimates in the text.

325: Residual currents should be total currents I believe... At least there are total currents shown in figure 10.

**A:** The sentence correctly identifies these as residual currents.

374: I think the wording is confusing here. Strongest shear is visible during summer, not early winter. What is visible is a deepening of the layer with maximum shear.

**A:** We edited the sentence.

376-377: This is visible for 2022, but in 2023, the maximum remains between 60-80 m. **A:** Ventilation reached >100m in 2023 as well – see yellow color at these depths in March-April of 2023.

392: I can see that the figure is in agreement with the point being made, but since it shows neither wind nor stratification I wonder how it can provide evidence for their role.

**A:** In the text, we referenced our earlier papers that argued for stronger stratification in the earlier years.

426-427: Is there any evidence for this?

**A:** We added a reference to Fig. 8, which compares current speed and shear between 2004-2009 and more recent years.

432-422: The relationship was already very strong before (-0.74); changes are probably hardly significant in the statistics.

**A:** This is not just a statistical observation; visual inspection clearly shows a tighter connection between SIC and |U|/Uz in recent years compared to the past.

Figure 5: Why is the wind time series split at 1991-2006 and 2007-2022? **A:** As indicated in our response above, the results are robust regardless of the averaging interval, with 2006/2007 used to split the 1991-2022 period into two equal parts.

Figure 7: In figure 6 there are correlation coefficients between ice and speed/shear; why not add them to figure 7 as well (instead of just mentioning those for shear in the text)?

A: Please see our response to the comment to line 314.

Figure 8: Name M1 should be changed to MB1 A: We removed this name from the figure.

Figure 12: I don't think the wavelet analysis adds anything to the story; it can be removed without loss.

**A:** The wavelet clearly highlights periods dominated by positive and negative correlations (see Reviewer's question MC4); therefore, we prefer to retain this panel.

Figure S2: X axis of b does not show correctly in the pdf.

A: We have corrected that – thank you.